# Two Pairs of 7,7′-Cyclolignan Enantiomers with Anti-Inflammatory Activities from *Perilla frutescens*

**DOI:** 10.3390/molecules27186102

**Published:** 2022-09-18

**Authors:** Jing Zuo, Tian-Hao Zhang, Liang Xiong, Lu Huang, Cheng Peng, Qin-Mei Zhou, Ou Dai

**Affiliations:** 1State Key Laboratory of Southwestern Chinese Medicine Resources, School of Pharmacy, Chengdu University of Traditional Chinese Medicine, Chengdu 611137, China; 2School of Pharmacy, Chengdu University of Traditional Chinese Medicine, Chengdu 611137, China; 3Institute of Innovative Medicine Ingredients of Southwest Specialty Medicinal Materials, School of Pharmacy, Chengdu University of Traditional Chinese Medicine, Chengdu 611137, China; 4Innovative Institute of Chinese Medicine and Pharmacy, Chengdu University of Traditional Chinese Medicine, Chengdu 611137, China

**Keywords:** *Perilla frutescens*, cyclolignans, enantiomers, anti-inflammatory activity, RAW 264.7 macrophages

## Abstract

*Perilla frutescens* (L.) Britt. (Labiatae), a medicinal plant, has been widely used for the therapy of multiple diseases since about 1800 years ago. It has been demonstrated that the extracts of *P. frutescens* exert significant anti-inflammatory effects. In this research, two pairs of 7,7′-cyclolignan enantiomers, possessing a cyclobutane moiety, (+)/(−)-perfrancin [(+)/(−)-**1**] and (+)/(−)-magnosalin [(+)/(−)-**2**], were separated from *P. frutescens* leaves. The present study achieved the chiral separation and determined the absolute configuration of (±)-**1** and (±)-**2**. Compounds (+)-**1** and (−)-**1** have notable anti-inflammatory effects by reducing the secretion of pro-inflammatory factors (NO, TNF-α and IL-6) and the expression of pro-inflammatory mediators (iNOS and COX-2). These findings indicate that cyclolignans are effective substances of *P. frutescens* with anti-inflammatory activity. The present study partially elucidates the mechanisms underlying the effects of *P. frutescens*.

## 1. Introduction

Chirality is very important in pharmacological research and drug development [1]. Drugs are mainly derived from natural products or by chemical synthesis, and, hence, often mixtures of several isomers are produced. Racemates are two enantiomers that exhibit identical physical and chemical properties, except for optical activities. As chiral separation techniques advance rapidly, people come to realize that some enantiomers may possess different properties with respect to pharmacological effects, toxicity and pharmacokinetics [2,3,4]. Some chiral medicines even exert opposite effects. For instance, (+)-picenadol is a potent opiate agonist, while (−)-picenadol acts as an opioid antagonist [5]. Therefore, it is important to study the enantiomers of chiral substances.

*Perilla frutescens* (L.) Britton (Labiatae) is a perennial herb that is extensively dispersed and cultivated not only in China, but also in other Asian countries [6]. As a well-known traditional edible and medicinal herb, *P. frutescens* has been used to treat common colds with cough and nausea and food poisoning such as fish or crab to humans [7]. Recent studies have demonstrated that about 90 compounds are derived from *P. frutescens*, including terpenoids, lignans and flavonoids [8]. Most of them have various pharmacological effects, including anti-microbial [9], anti-allergic [10], anti-inflammatory [11], antioxidant [12] and anti-depressive activities [13]. As part of our systematic study of bioactive products from traditional Chinese medicines, the isolation, structural elucidation and anti-inflammatory activity evaluation of the isolated compounds from *P. frutescens* were explored in the current study.

## 2. Results

### 2.1. Structural Elucidation of the Isolated Compounds

Compound **1** was obtained as colorless oil. Its molecular formula (C_24_H_32_O_6_) is indicated by an HR-ESI-MS ion peak at *m/z* 439.2099 [M + Na]^+^ (calcd. for C_24_H_32_O_6_Na, 439.2097), which signifies that it has nine degrees of unsaturation. The IR spectrum of **1** exhibits characteristic absorption bands for aromatic (1608, 1511 and 1453 cm^−1^) functionalities. The ^1^H NMR spectrum of **1** shows signals attributed to two 1,2,4,5-tetrasubstituted aromatic rings [*δ*_H_ 6.30 (1H, s), 6.38 (1H, s), 6.44 (1H, s) and 6.80 (1H, brs)], six aromatic methoxys [*δ*_H_ 3.37 (3H, s), 3.39 (3H, s), 3.75 (3H, s), 3.80 (3H, s), 3.80 (3H, s) and 3.84 (3H, s)], four methines [*δ*_H_ 2.39 (1H, m), 2.49 (1H, m), 3.55 (1H, t, *J* = 9.6 Hz) and 4.40 (1H, t, *J* = 9.6 Hz)] and two methyls [0.66 (3H, d, *J* = 7.2 Hz) and 1.28 (3H, d, *J* = 6.6 Hz)] (Table 1). The ^13^C NMR and DEPT data reveal the presence of twenty-four carbons assignable to the above-protonated units and eight aromatic quaternary carbons (six of which were oxygenated). By the analysis of ^1^H and ^13^C NMR data, compound **1** was identified as a new natural lignin (Figure 1), which was only reported as a by-product (perfrancin) in the process of synthesizing magnoshinin [14]. The compound’s absolute configuration has not been determined.

Further analysis of the HSQC, ^1^H,^1^H-COSY, HMBC and NOESY data confirm the planar structure and relative configuration of compound **1**. Interestingly, no cotton effects are observed in its ECD spectrum, which indicates that **1** might be isolated as a racemic mixture. Subsequently, compound **1** was further enantioseparated on a chiral column to afford (+)-**1** and (−)-**1**. Their absolute configurations were identified as (+)-(7*S*,8*S*,7′*R*,8′*S*)-**1** and (−)-(7*R*,8*R*,7′*S*,8′*R*)-**1,** based on experimental and calculated ECD data (Figure 2 and Appendix A). Enantiomers are labeled (−)-perfrancin and (+)-perfrancin.

Compound **2** was obtained as a colorless crystal. It has a molecular formula of C_24_H_32_O_6_, as indicated by HRESIMS at *m/z* 439.2092 [M + Na]^+^ (calcd. for C_24_H_32_O_6_Na, 439.2097). The ^1^H and ^13^C NMR spectra of **2** display the same resonances as those of magnosalin, which was confirmed to have a small negative optical rotation {${[\alpha ]\;}_{D}^{20}$ −4.64 (*c* 0.25, CHCl_3_)} [11]. Moreover, magnosalin’s absolute configuration has not been ascertained. Interestingly, X-ray diffraction data analysis of compound **2** (Figure 3) suggests that it contains two racemates. The (+)-**2** and (−)-**2** enantiomers successfully obtained by chiral HPLC separation display obvious specific optical rotation {(−)-**2**: ${[\alpha ]\;}_{D}^{20}$ –21.2 (*c* 0.10, MeOH), −33.8 (*c* 0.07, CHCl_3_); (+)-**2**: +23.0 (*c* 0.05, MeOH), +37.4 (*c* 0.05, CHCl_3_)}. Their absolute configurations were determined via ECD calculations. As shown in Figure 2, the absolute configuration of (+)-**2** is identified as 7*R*,8*S*,7′*R*,8′*S* based on the good agreement with the calculated ECD spectrum of (7*R*,8*S*,7′*R*,8′*S*)-**2**, naturally elucidating the absolute configuration of (−)-**2** as (7*S*,8*R*,7′*S*,8′*R*)-**2**. Consequently, the latter configuration (7*S*,8*R*,7′*S*,8′*R*) is clarified as the reported compound (−)-magnosalin [11], whereas the former one (7*R*,8*S*,7′*R*,8′*S*) corresponds to an unreported compound labeled (+)-magnosalin.

### 2.2. Physicochemical Properties and Spectroscopic Data of Compounds **1** and **2**

Perfrancin (**1**): colorless oil; UV (MeCN) *λ*_max_ (log *ε*) 201 (4.41), 232 (3.80), 293 (3.50) nm; IR (ATR) *ν*_max_ 2949, 2864, 1608, 1511, 1453, 1397, 1322, 1260, 1202, 1031, 853, 817, 800 cm^−1^; (+)-HR-ESI-MS *m*/*z* 439.2099 [M + Na]^+^ (calcd. for C_24_H_32_O_6_Na, 439.2097). Table 1 shows ^1^H and ^13^C NMR data. The (+)-(7*S*,8*S*,7′*R*,8′*S*)-**1**: ${[\alpha ]\;}_{D}^{20}\;$+15.2 (*c* 0.05, MeOH); ECD (MeCN) *λ*_max_ (Δ*ε*) 219 (−13.70), 245 (+13.87), 265 (+2.42), 294 (+8.96) nm. The (−)-(7*R*,8*R*,7′*S*,8′*R*)-**1**: ${[\alpha ]\;}_{D}^{20}$ −21.2 (*c* 0.10, MeOH); ECD (MeCN) *λ*_max_ (Δ*ε*) 219 (+17.25), 240 (−13.10), 265 (−1.85), 293 (−7.92) nm.

Magnosalin (**2**): colorless crystals (MeOH); UV (MeCN) *λ*_max_ (log *ε*) 203 (4.10), 233 (3.50), 295 (3.24) nm; IR (ATR) *ν*_max_ 2950, 2899, 2831, 1610, 1519, 1462, 1392, 1313, 1271, 1205, 1035, 858, 820, 755, 683 cm^−1^; (+)-HR-ESI-MS *m*/*z* 439.2092 [M + Na]^+^ (calcd. for C_24_H_32_O_6_Na, 439.2097). Table 1 shows ^1^H and ^13^C NMR data. The (+)-(7*R*,8*S*,7′*R*,8′*S*)-**2**: ${[\alpha ]\;}_{D}^{20}$+23.0 (*c* 0.16, MeOH), +37.4 (*c* 0.05, CHCl_3_); ECD (MeCN) *λ*_max_ (Δ*ε*) 194 (−7.16), 206 (+7.24), 218 (+0.97), 228 (+2.68), 244 (−2.20), 300 (+0.90) nm. The (−)-(7*S*,8*R*,7′*S*,8′*R*)-**2**: ${[\alpha ]\;}_{D}^{20}\;$−21.2 (*c* 0.10, MeOH), −33.8 (*c* 0.07, CHCl_3_); ECD (MeCN) *λ*_max_ (Δ*ε*) 193 (+5.93), 206 (−4.52), 218 (−1.31), 228 (−2.94), 244 (+2.40), 301 (−0.99) nm.

**X-ray Crystallographic Analysis.** Crystals of **2** were obtained from MeOH (Figure 3). We measured intensity data on a Bruker D8 Quest diffractometer equipped with an APEX-II CCD using Cu K*α* radiation. Crystal data for **2**: C_24_H_32_O_6_, M = 416.49, monoclinic, *a* = 5.9842(8) Å, *b* = 17.735(3) Å, *c* = 22.041(3) Å, *α* = 90°, *β* = 91.600(9)°, *γ* = 90°, *V* = 2338.3(6) Å^3^, space group *P*2_1_/n, *T* = 293(2) K, *Z* = 4, *μ*(Cu K*α*) = 0.685 mm^−1^, 23,067 reflections measured, 4,192 independent reflections (*R*_int_ = 0.0690), average redundancy 5.503, completeness = 97.6%. Final *R* indices (*I* > *2σ*(*I*)): *R*_1_ = 0.0549, *wR*_2_ = 0.1224. Final *R* indices (all data): *R*_1_ = 0.0985, *wR*_2_ = 0.1458. The goodness-of-fit on *F*^2^ was 1.094. CCDC number: 2191989.

### 2.3. Effects of Compounds (+)-**1** and (−)-**1** on Cell Viability of RAW 264.7 Macrophages

Cytotoxicity of compounds (+)-**1** and (−)-**1** was investigated by measuring cell viability. As shown in Figure 4, compounds (+)-**1** and (−)-**1** exert no cytotoxicity on RAW 264.7 macrophages at 3.13, 6.25, 12.5, 25 and 50 μM. Therefore, compounds (+)-**1** and (−)-**1** at 3.13–50 μM were used in subsequent experiments.

### 2.4. Effects of Compounds (+)-**1** and (−)-**1** on NO Production in RAW 264.7 Macrophages

NO has been considered an important mediator whose production increases in the case of inflammation [15]. Therefore, inhibition of excessive NO production is commonly used to assess anti-inflammatory effects of compounds [16]. Levels of NO production in the supernatant of RAW 264.7 macrophages activated by LPS were investigated using the Griess reagent Kits. Figure 5 shows that, in comparison with the control group, NO production was higher in the LPS-treated group. Administration of (+)-**1** or (−)-**1** inhibits NO synthesis in RAW 264.7 macrophages was activated by LPS in a dose-dependent manner. The IC_50_ values of (+)-**1** and (−)-**1** are 21.61 ± 2.35 and 28.02 ± 1.93 μM, respectively. In addition, we used curcumin as a positive control (IC_50_ = 20.37 ± 0.77 μM).

### 2.5. Effects of Compound (+)-**1** on TNF-α and IL-6 Production in RAW 264.7 Macrophages

Macrophages were cultured in the presence of compound (+)-**1** and LPS to further characterize the anti-inflammatory effects of compound (+)-**1**. The concentrations of TNF-α (*p* < 0.01, Figure 6A) and IL-6 (*p* < 0.01, Figure 6B) rose to high levels following treatment with LPS alone, while compound (+)-**1** at 12.5, 25 and 50 μM markedly inhibited their release into the supernatant (*p* < 0.01).

### 2.6. Effects of Compound (+)-**1** on iNOS and COX-2 Protein Expression in RAW 264.7 Macrophages

Many critical enzymes are related to the establishment and progression of inflammation, including iNOS and COX-2 [17,18]. We carried out Western blot analysis on LPS-treated RAW 264.7 macrophages with or without compound (+)-**1** to determine protein expression levels. LPS treatment at 1 µg/mL for 24 h markedly increased iNOS (*p* < 0.01, Figure 7A and Appendix A) and COX-2 (*p* < 0.01, Figure 7B and Appendix A) protein expression, in comparison with the control group. Compound (+)-**1** treatment significantly reversed the increased expression of iNOS (*p* < 0.01) and COX-2 (*p* < 0.01) at 50 μM in RAW 264.7 macrophages activated by LPS.

## 3. Discussion

As lignans, no more than 200 7,7′-cyclolignans have been identified in nature, which are mainly reported from the Labiatae [19,20,21], Asteraceae [22,23], Piperaceae [24,25] and Ginkgoaceae families [26]. In total, Labiatae accounts for less than 30 compounds. Two pairs of 7,7′-cyclolignan enantiomers, (±)-perfrancin (**1**) and (±)-magnosalin (**2**), were separated from the leaves of *P. frutescens*. Perfrancin has been reported only once before in organic synthesis [14], and this is the first report of perfrancin from natural plants. Magnosalin has been isolated from *Magnolia salicifolia* [27], *Piper sumatranum* [24] and *P. frutescens* [11]. However, absolute configurations of these compounds have not been determined. In the current study, we found that two pairs of lignan enantiomers were further separated through chiral HPLC separation, and compounds’ absolute configurations were successfully determined by X-ray diffraction analyses and ECD calculations.

As an essential inflammatory mediator, NO is overproduced in macrophages stimulated by LPS [28]. As reported before, compound **2** is regarded as an inhibitor of NO synthase [11]. In our present study, 7,7′-cyclolignan enantiomers (+)-**1** and (−)-**1** were assessed for their anti-inflammatory effect against the release of NO in RAW 264.7 macrophages stimulated by LPS. They have similar inhibitory effects on NO production. In acute and chronic diseases, TNF-α is an essential factor in the inflammatory reaction [29]. Another cytokine, IL-6, also plays a critical role in inflammation [30,31]. Thus, a therapeutic strategy to reduce inflammation could be suppressing the production of inflammatory mediators and inhibiting the release of pro-inflammatory cytokines. In the present study, LPS treatment significantly increased the secretion of TNF-α and IL-6, which was markedly suppressed by compound (+)-**1**.

Inflammatory processes are initiated and sustained by iNOS and COX-2, which play key roles in the production of NO [32,33]. Many studies have confirmed that the expression of iNOS and COX-2 is highly upregulated during infection [34,35,36]. Our Western blot results revealed that the excessive expression of iNOS and COX-2 induced by LPS in RAW 264.7 macrophages was significantly inhibited by compound (+)-**1**. Thus, our findings provide further evidence that compound (+)-**1** has anti-inflammatory activity.

## 4. Materials and Methods

### 4.1. Plant Material

We obtained *P. frutescens* leaves from Sichuan Neautus Traditional Chinese Medicine Co., Ltd. (Chengdu, China). Leaves were identified by Dr. Jihai Gao (Chengdu University of TCM, Chengdu, China). A voucher specimen (ZS-20171215) was deposited at the State Key Laboratory of Southwestern Chinese Medicine Resources, Chengdu University of TCM.

### 4.2. Extraction and Isolation

All chemical materials are shown in Appendix A. After air-drying the *P. frutescens* leaves (10 kg), we extracted them with 95% EtOH (3 × 50 L) for 3 × 2 h under reflux. A yellow residue (2.1 kg) was obtained after evaporating the EtOH extract under reduced pressure, which was suspended in H_2_O and partitioned with EtOAc. With a gradient elution of petroleum ether–EtOAc (100:1–0:1, *v*/*v*), 18 fractions (Fr.1–Fr.18) were obtained from the EtOAc extract (620 g). Three subfractions (Fr.15-1–Fr.15-3) were obtained from fraction Fr.15 (16.4 g) after chromatographic separation over a polyamide column. A Sephadex LH-20 column was used for chromatography to separate Fr.15-1 into three subfractions (Fr.15-1a–Fr.15-1c).

Successive purification of Fr.15-1a with silica gel column chromatography (petroleum ether–Et_2_O, 8:1), PTLC (petroleum ether–EtOAc, 3:1) and reversed-phase semipreparative HPLC (74% MeOH in H_2_O) yielded **1** (12.5 mg, *t*_R_ = 125 min) and **2** (2.4 mg, *t*_R_ = 140 min). Racemic compounds **1** and **2** were chirally separated through a Chiralpak IG column (95% MeOH in H_2_O) to afford (−)-**1** (5.4 mg, *t*_R_ = 4.9 min)/(+)-**1** (4.5 mg, *t*_R_ = 6.0 min) and (−)-**2** (0.9 mg, *t*_R_ = 7.4 min)/(+)-**2** (1.1 mg, *t*_R_ = 16.1 min), respectively.

### 4.3. Cell Culture

RAW 264.7 macrophages (CL-0190, Procell, Wuhan, China) were incubated in DMEM (C11995500BT, Gibco, Grand Island, NY, USA) containing 10% FBS (11011-8611, Tianhang, Huzhou, China), 100 U/mL penicillin and 100 µg/mL streptomycin (BL505A, Biosharp, Beijing, China). Cells were cultured in a humidified environment with 5% CO_2_ at 37 °C.

### 4.4. Cell Viability Assay

Compounds (+)-**1** and (−)-**1** were tested for cytotoxicity using MTT (EZ7890B315, BioFroxx, GER) assays. In brief, we plated macrophages in 96-well plates (5 × 10^4^ per well) and cultivated them for 24 h before treatment with the test compounds. Then, macrophages were incubated with compounds (+)-**1** and (−)-**1** (3.13, 6.25, 12.5, 25 or 50 µM) for 24 h. After adding 20 μL MTT (5 mg/mL), 150 μL DMSO (PYG0040, Boster, Wuhan, China) were added to dissolve the formazan crystals. An Analytical Microplate Reader (Thermo Fisher Scientific, Waltham, MA, US) was used to measure the absorbance at 490 nm.

### 4.5. NO Production Assay

Compounds (+)-**1** and (−)-**1** were tested for inhibition of NO production using the Griess reagent Kit (S0021S, Beyotime, Shanghai, China). Briefly, we plated macrophages in 96-well plates (5 × 10^4^ per well) and cultured them for 24 h. In the model group, macrophages were incubated with LPS (1 µg/mL), while in the drug group, macrophages were incubated with compounds (+)-**1** and (−)-**1** (3.13, 6.25, 12.5, 25, or 50 µM) and LPS (1 µg/mL). After incubation for 24 h, the NO concentration in the culture medium was determined by measuring the optical density at 540 nm following the manufacturer’s instructions.

### 4.6. Cytokines Production Assays

Compound (+)-**1** was tested for inhibition of TNF-α and IL-6 production using ELISA kits (E-EL-M0049c, E-EL-M0044c, Elabscience, Wuhan, China). Briefly, we plated macrophages in 24-well plates (2.5 × 10^5^ per well) and cultured them for 24 h. In the model group, macrophages were incubated with LPS (1 µg/mL); while in the drug group, macrophages were incubated with compound (+)-**1** (12.5, 25, or 50 µM) and LPS (1 µg/mL). After incubation for 24 h, the TNF-α and IL-6 concentrations in the culture medium were determined by measuring the optical density at 450 nm following the manufacturer’s instructions.

### 4.7. Western Blot Analysis

RAW 264.7 macrophages were seeded on 6-centimeter dishes (5 × 10^5^ per dish) and cultured for 24 h. In the model group, macrophages were incubated with LPS (1 µg/mL); while in the drug group, macrophages were incubated with compound (+)-**1** (12.5, 25 or 50 µM) and LPS (1 µg/mL). The next day, the macrophages were rinsed with precooled PBS (C10010500BT, Gibco, Grand Island, NY, USA) and lysed with RIPA buffer (P0013, Beyotime, Shanghai, China) for 30 min on ice. After centrifugation at 12,000× *g* for 15 min, the protein concentration was determined using the BCA assay (P0010, Beyotime, Shanghai, China). Proteins (30 µg per lane) were separated by SDS-PAGE (03659300, EpiZyme, Shanghai, China) and transferred to PVDF membranes (IPVH00010, Millipore Sigma, Burlington, Massachusetts, USA). After blocking with 5% skimmed milk, the membranes were incubated overnight with anti-β-actin (340042, Zen Bio, Chengdu, China), anti-iNOS (AF7281, Beyotime, Shanghai, China) and anti-COX-2 (AF1924, Beyotime, Shanghai, China) at 4 °C. After washing three times in TBST, membranes were incubated with HRP-conjugated antibody (511203, Zen Bio, Chengdu, China) for 2 h at 37 °C. After three washes with TBST, Tanon 5200 chemiluminescent imaging (Tanon, China) was used to detect proteins of interest.

### 4.8. Statistical Analysis

Each experiment was conducted three times. Data are presented as the mean with SEM. Multiple groups were analyzed by ANOVA combined with Tukey’s post hoc test. Statistical significance was defined as *p* < 0.05.

## 5. Conclusions

In this study, two pairs of 7,7′-cyclolignan enantiomers, (+)/(−)-perfrancin [(+)/(−)-**1**] and (+)/(−)-magnosalin [(+)/(−)-**2**], were separated by chiral separation, and their absolute configurations were determined. In RAW 264.7 macrophages activated by LPS, compounds (+)-**1** and (–)-**1** markedly suppressed nitric oxide production. Further investigations on compound (+)-**1** indicated that it exerts anti-inflammatory effects via suppressing the production of TNF-α and IL-6 and protein expression of iNOS and COX-2 in RAW 264.7 macrophages stimulated by LPS.

## Figures and Tables

**Figure 1 molecules-27-06102-f001:**
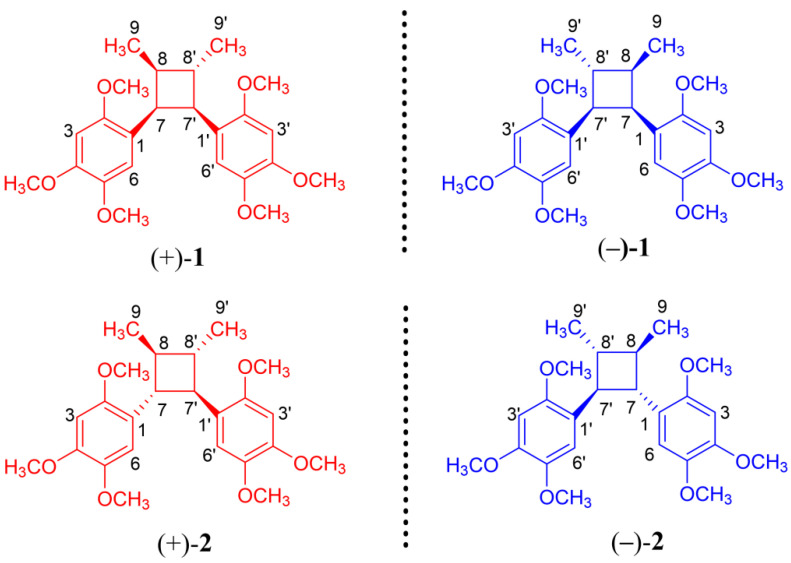
Structures of compounds **1** and **2**.

**Figure 2 molecules-27-06102-f002:**
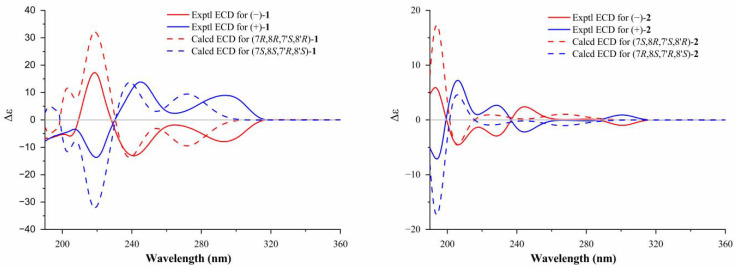
Experimental and calculated ECD spectra of **1** and **2**.

**Figure 3 molecules-27-06102-f003:**
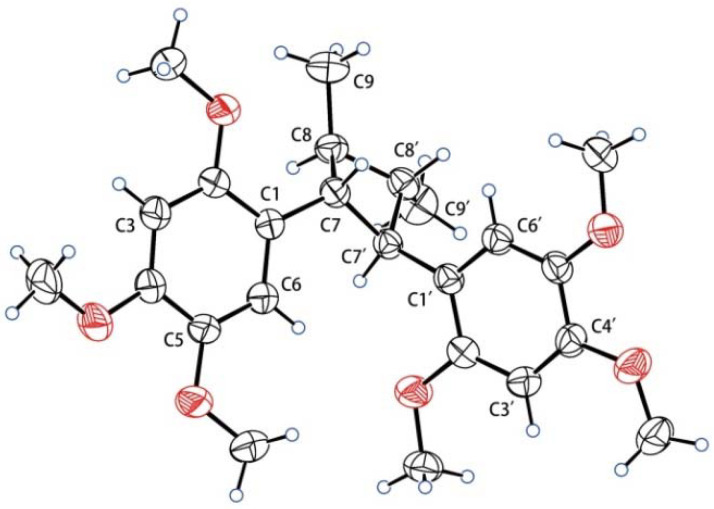
X-ray crystallographic structure of **2**.

**Figure 4 molecules-27-06102-f004:**
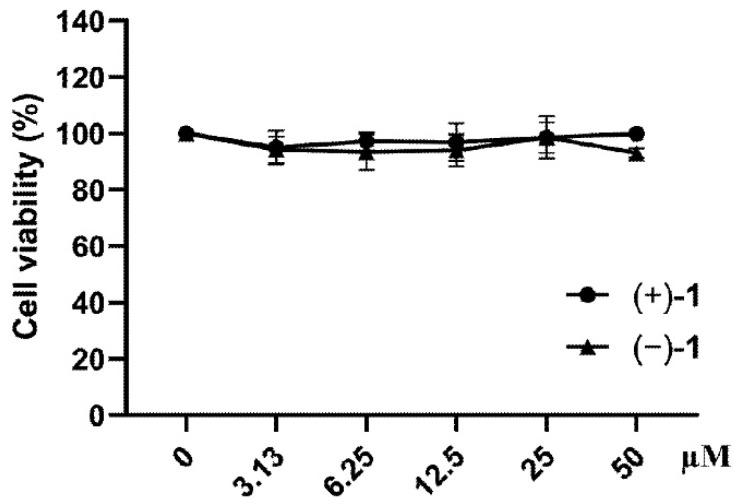
Compounds (+)-**1** and (−)-**1** had no effect on RAW 264.7 cells. Values are presented as mean ± SEM of three independent experiments.

**Figure 5 molecules-27-06102-f005:**
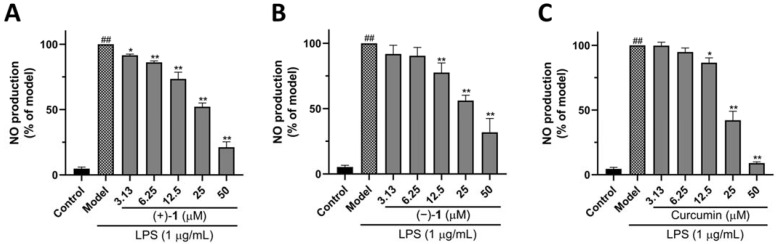
The inhibitory effect of compounds (+)-**1** and (−)-**1** against LPS-induced NO production on RAW 264.7 cells. (**A**) Compound (+)-**1** reduced the production of NO. (**B**) Compound (−)-**1** reduced the production of NO. (**C**) Curcumin reduced the production of NO. Values are presented as mean ± SEM of three independent experiments. ^##^
*p* < 0.01 vs. control group, * *p* < 0.05, ** *p* < 0.01 vs. model group.

**Figure 6 molecules-27-06102-f006:**
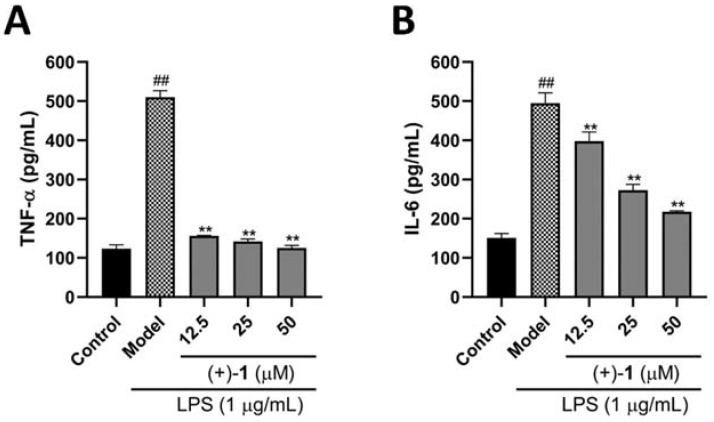
Effects of compound (+)-**1** on inflammation-related cytokines on RAW 264.7 cells. (**A**) Compound (+)-**1** decreased TNF-α expression levels. (**B**) Compound (+)-**1** decreased IL-6 expression levels. Values are presented as mean ± SEM of three independent experiments. ^##^
*p* < 0.01 vs. control group, ** *p* < 0.01 vs. model group.

**Figure 7 molecules-27-06102-f007:**
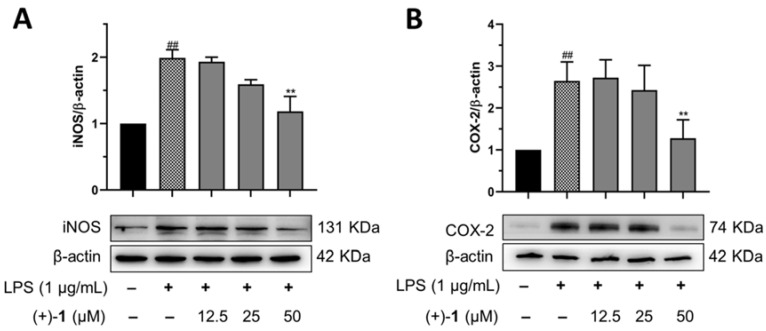
Effects of compound (+)-**1** on inflammation-related protein expression on RAW 264.7 cells. (**A**) Compound (+)-**1** decreased iNOS expression levels. (**B**) Compound (+)-**1** decreased COX-2 expression levels. Expression ratios of data are shown as mean ± SEM of three independent experiments and normalized with β-actin. ^##^
*p* < 0.01 vs. control group, ** *p* < 0.01 vs. model group.

**Table 1 molecules-27-06102-t001:** The ^1^H (600 MHz) and ^13^C NMR (150 MHz) data of compounds **1** and **2** in CDCl_3_ (*δ* in ppm, *J* in Hz).

Position	Compound 1		Compound 2	
	*δ* _H_	*δ* _C_	*δ* _H_	*δ* _C_
1		121.0		124.0
2		152.4		151.8
3	6.44 s	98.0	6.46 s	98.0
4		146.5		147.7
5		141.8		143.2
6	6.38 s	113.5	6.95 s	112.3
7	4.40 t (9.6)	39.2	3.26 t (overlapped)	45.5
8	2.39 m	39.6	1.76 m	43.6
9	0.66 d (7.2)	15.1	1.18 d (6.0)	19.2
1′		122.5		124.0
2′		152.0		151.8
3′	6.30 s	98.2	6.46 s	98.0
4′		147.9		147.7
5′		143.1		143.2
6′	6.80 brs	111.5	6.95 s	112.3
7′	3.55 t (9.6)	44.2	3.26 t (overlapped)	45.5
8′	2.49 m	41.3	1.76 m	43.6
9′	1.28 d (6.6)	20.1	1.18 d (6.0)	19.2
OMe-2	3.75 s	57.3	3.69 s	56.7
OMe-4	3.80 s	56.1	3.85 s	56.3
OMe-5	3.37 s	56.1	3.87 s	56.9
OMe-2′	3.39 s	56.0	3.69 s	56.7
OMe-4′	3.80 s	56.3	3.85 s	56.3
OMe-5′	3.84 s	57.2	3.87 s	56.9

## Data Availability

The data presented in this study are available in the Appendix A or can be provided by the authors.

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
