# Peer review of "Two Pairs of 7,7′-Cyclolignan Enantiomers with Anti-Inflammatory Activities from *Perilla frutescens"

_molecules, 2022, doi:10.3390/molecules27186102_

Round 1

Reviewer 1 Report

This manuscript reports the discovery of unusual 7,7′-cyclolignan enantiomers possessing a cyclobutane moietyCompounds (+)-1 and (-)-1 have notable anti-inflammatory effects. The article is clear and the structural analyses are reliable. From my point of view, this manuscript would be suitable for publication in Molecules with some minor revisions.

1. The spliting peak of H-7 and H-7′ should be double peak.

2. The 13C NMR spectrum does not need to be integrated in the Supporting Information.

Author Response

Point 1:  The spliting peak of H-7 and H-7ʹ should be double peak.

Response 1: Thanks for your comment. The splitting peak of H-7 and H-7ʹ should be a triple peak because H-7 has vicinal coupling effects on H-8 and H-7ʹ; H-7ʹ has vicinal coupling effects on H-7 and H-8ʹ, and none of their dihedral angles is close to 90 degrees. In addition, the peaks of H-7 [δH 3.26 (1H, t)] and H-7ʹ [δH 3.26 (1H, t)] of compounds (±)-2 are overlapped, so it is difficult to calculate the exact coupling constants.

Point 2: The13C NMR spectrum does not need to be integrated in the Supporting Information.

Response 2: Thanks for your good suggestion. We have revised them in the Supporting Information.

Reviewer 2 Report

Overall Zuo, et. al have done a nice job in isolating a new 7',7'-cylolignan cyclobutane natural product and have done a good job in characterizing the structural attributes and have done an initial job in characterizing some modest anti-inflammatory activities. 

There are several instances where the novelty is oversold i.e. "rare", "outstanding", "first" and the reviewer would advise to revise the adjectives.  The team has done a good initial job in trying to profile the biological activity in connection with the medicinal properties of the isolation plant.  The anti-inflammatory activities are modest, mid uM levels in generic assays.  

I'm a bit concerned with the abrupt jump from pharmacological activity characterization to the modeling/docking done with GR.  I think the modeling and docking have been done at a rather basic level and draw too strong of conclusions from the author that the modest anti-inflammatory activity is directly correlated to GR.  For publication there needs to be a better direct connection with at minimum some genetic or biophysical connection between the newly identified compounds and GR or this docking section should be completely removed. 

Line 17-18- clarify “ancient times”, too unspecific.

Line 19 – omit “outstanding” – meaningless and unqualified adjective

Line 19 – omit “rare” unsubstantiated

Line 21 – omit “first”

Line 45 – by “fish and crab poisoning” is this seafood/food poisoning in humans or toxicity to the fish/crab? Please clarify

Line 181- The jump to GR seems to be strange. There is also precedent in cyclobutane compounds are androgen receptor (AR) antagonists (see: Inhibiting androgen receptor nuclear entry in castration-resistant prostate cancer | Nature Chemical Biology).

Line 193- “This result indicates that the four-memebered ring is important for the potent GR activity”  This is an assumption as there is no data that challenges this. I’d suggest rephrasing/reformatting.

Line 211-  with 200 analogues I wouldn’t necessarily classify as “unusual”

Section 2.7 Molecular docking to GR.

  Overall this section is unconvincing in connecting the structure of the 7,7’-cylolignan natural products to their antiinflammatory mechanism.  Further genetic or biophysical data needs to support this to enable inclusion of the modeling/docking for publication

Author Response

Point 1 There are several instances where the novelty is oversold i.e. "rare", "outstanding", "first" and the reviewer would advise to revise the adjectives. The team has done a good initial job in trying to profile the biological activity in connection with the medicinal properties of the isolation plant. The anti-inflammatory activities are modest, mid μM levels in generic assays.

Response 1: Thanks for your good advice. We have revised these adjectives in the manuscript.

Point 2 I'm a bit concerned with the abrupt jump from pharmacological activity characterization to the modeling/docking done with GR. I think the modeling and docking have been done at a rather basic level and draw too strong of conclusions from the author that the modest anti-inflammatory activity is directly correlated to GR. For publication there needs to be a better direct connection with at minimum some genetic or biophysical connection between the newly identified compounds and GR or this docking section should be completely removed.

Response 2: Thanks for your helpful suggestion. Reference (see: Endiandrin A, a potent glucocorticoid receptor binder isolated from the Australian plant Endiandra anthropophagorum) verifies that the 7,7ʹ-cylolignan natural products can bind very well with GR by Fluorescence Polarization (FP) experiments, which was similar to the results of molecular docking in this manuscript. However, due to the insufficient amount of compounds (±)-1, we could not do further experiments to verify this result. Thus, we accept your advice to delete “Section 2.7 Molecular docking to GR” in the manuscript.

Point 3 Line 17-18- clarify “ancient times”, too unspecific.

Response 3: Thanks for your kindly and helpful suggestion. We have changed the “ancient times” into “about 1800 years ago”.

Point 4 Line 19 – omit “outstanding” – meaningless and unqualified adjective.

Response 4: Thanks for your good suggestion. We have changed “outstanding” into “significant” in order to be more objective.

Point 5 Line 19 – omit “rare” unsubstantiated.

Response 5: Thanks for your good suggestion. We have omitted “rare”.

Point 6 Line 21 – omit “first”.

Response 6: Thanks for your good suggestion. We have omitted “first”.

Point 7 Line 45 – by “fish and crab poisoning” is this seafood/food poisoning in humans or toxicity to the fish/crab? Please clarify.

Response 7: Thanks for your careful review work. “fish and crab poisoning” is the seafood/food poisoning in humans. The description in the article may not be clear, we have revised them.

Point 8 Line 181 – The jump to GR seems to be strange. There is also precedent in cyclobutane compounds are androgen receptor (AR) antagonists (see: Inhibiting androgen receptor nuclear entry in castration-resistant prostate cancer | Nature Chemical Biology).

Response 8: Thanks for your helpful suggestion. Due to the insufficient amount of compounds (±)-1, we could not do further experiments to verify this conclusion. Thus, we have deleted “Section 2.7 Molecular docking to GR” in the manuscript.

 Point 9 Line 193 – “This result indicates that the four-memebered ring is important for the potent GR activity” This is an assumption as there is no data that challenges this. I’d suggest rephrasing/reformatting.

Response 9: Thanks for your helpful suggestion. We have deleted “Section 2.7 Molecular docking to GR” in the manuscript.

Point 10 Line 211- with 200 analogues I wouldn’t necessarily classify as “unusual”.

Response 10: Thanks for your helpful suggestion. We have revised them in the manuscript.

Point 11 Section 2.7 Molecular docking to GR.

Overall this section is unconvincing in connecting the structure of the 7,7ʹ-cylolignan natural products to their anti-inflammatory mechanism. Further genetic or biophysical data needs to support this to enable inclusion of the modeling/docking for publication.

Response 11: Thanks for your helpful suggestion. We have deleted “Section 2.7 Molecular docking to GR” in the manuscript.

Reviewer 3 Report

The research is interesting and the writting of the manuscript is also relative good. If the structures of the new compounds were not described in the introduction could be better. The introduction need not introduce the results of the research. Additionally, the manuscript said a positve control was used, but  the experiment results were not shown on the figures.

Author Response

Point 1: The research is interesting and the writing of the manuscript is also relative good. If the structures of the new compounds were not described in the introduction could be better. The introduction need not introduce the results of the research. Additionally, the manuscript said a positive control was used, but the experiment results were not shown on the figures.

Response: Thanks for your kindly and helpful suggestion. The structures of the new compounds have been moved to the results part. And we have also revised the introduction. At the same time, we have added the positive control to the manuscript.

Round 2

Reviewer 2 Report

I appreciate the authors removing the modeling/docking results and clarifying and enhancing the rest of the publication.  This is an interesting cyclobutane natural product family and will be great to add the identification, characterization, and biological activity to the field.